# Prevalence and Associated Factors of Nocturnal Eating Behavior and Sleep-Related Eating Disorder-Like Behavior in Japanese Young Adults: Results of an Internet Survey Using Munich Parasomnia Screening

**DOI:** 10.3390/jcm9041243

**Published:** 2020-04-24

**Authors:** Kentaro Matsui, Yoko Komada, Katsuji Nishimura, Kenichi Kuriyama, Yuichi Inoue

**Affiliations:** 1Japan Somnology Center, Neuropsychiatric Research Institute, Tokyo 1510053, Japan; matsui.kentaro@ncnp.go.jp; 2Department of Psychiatry, Tokyo Women’s Medical University, Tokyo 1628666, Japan; nishimura.katsuji@twmu.ac.jp; 3Clinical Laboratory, National Institute of Mental Health, National Center of Neurology and Psychiatry, Tokyo 1878551, Japan; 4Department of Sleep-Wake Disorders, National Institute of Mental Health, National Center of Neurology and Psychiatry, Tokyo 1878551, Japan; kenichik@ncnp.go.jp; 5Liberal Arts, Meiji Pharmaceutical University, Tokyo 2048588, Japan; yoko.komada@gmail.com; 6Department of Somnology, Tokyo Medical University, Tokyo 1608402, Japan

**Keywords:** nocturnal eating syndrome, sleep-related eating disorder, eating disorder, parasomnia, delayed sleep-wake phase, MUPS

## Abstract

Nocturnal (night) eating syndrome and sleep-related eating disorder have common characteristics, but are considered to differ in their level of consciousness during eating behavior and recallability. To date, there have been no large population-based studies determining their similarities and differences. We conducted a cross-sectional web-based survey for Japanese young adults aged 19–25 years to identify factors associated with nocturnal eating behavior and sleep-related eating disorder-like behavior using Munich Parasomnia Screening and logistic regression. Of the 3347 participants, 160 (4.8%) reported experiencing nocturnal eating behavior and 73 (2.2%) reported experiencing sleep-related eating disorder-like behavior. Smoking (*p* < 0.05), use of hypnotic medications (*p* < 0.01), and previous and/or current sleepwalking (*p* < 0.001) were associated with both nocturnal eating behavior and sleep-related eating disorder-like behavior. A delayed sleep-wake schedule (*p* < 0.05) and sleep disturbance (*p* < 0.01) were associated with nocturnal eating behavior but not with sleep-related eating disorder-like behavior. Both nocturnal eating behavior and sleep-related eating disorder-like behavior had features consistent with eating disorders or parasomnias. Nocturnal eating behavior but not sleep-related eating disorder-like behavior was characterized by a sleep-awake phase delay, perhaps representing an underlying pathophysiology of nocturnal eating syndrome.

## 1. Introduction

Nocturnal (night) eating syndrome (NES) is characterized by recurrent episodes of eating after the evening meal or after awakening from sleep [1,2]. NES has been described as “other specified feeding or eating disorder” in the Diagnostic and Statistical Manual of Mental Disorders, Fifth Edition [1], but its criteria are relatively unclear. Therefore, the diagnostic criteria proposed by Allison and colleagues [2] have been also used. Similar eating behaviors, but with total amnesia or partial unawareness of eating or drinking, are seen in sleep-related eating disorder (SRED) [3,4,5]. SRED can be distinguished from NES by its unconscious eating behavior and its inability to recall, which was emphasized in the diagnostic criteria for SRED in the International Classification of Sleep Disorders, Third Edition (ICSD-3) [6]. However, since NES and SRED share several features, the distinction between the two is still controversial [7,8]. NES has been considered an eating disorder with evening hyperphagia, eating episodes occurring upon awakening during the night and resultant morning anorexia [9,10]; the pathology of NES is believed to be a delayed circadian pattern of food intake [11,12]. SRED is classified as parasomnia in the ICSD-3 [6] and characterized by eating while “half-awake, half-asleep” or “asleep” [4,6]. SRED patients often experience sleepwalking [3,4], which is rare in NES patients [13]. To date, no large population-based studies have been conducted on NES and SRED [8], and only a few studies have addressed their similarities and differences in clinical characteristics and pathophysiology [7,13].

Munich Parasomnia Screening (MUPS) is a self-assessment questionnaire consisting of 21 items and is used to evaluate parasomnias and sleep-related movement disorders, including NES and SRED [14]. The questionnaire covers past history and current frequency of abnormal nighttime behaviors and their sensitivity and specificity [14]. Our group previously translated and validated a Japanese version of MUPS [15]. With this questionnaire, we conducted an internet survey for Japanese young adults from the general population and estimated the prevalence of NES and SRED in this cohort. We then analyzed factors related to nocturnal eating behavior and SRED-like behavior in this population to determine similarities and differences in these two disorders.

## 2. Experimental Section

The present study was conducted as part of a comprehensive research project on the influence of sleep schedule on daytime functioning and depression among young adults [16,17]. This cross-sectional web-based questionnaire for Japanese individuals aged 19–25 years was conducted in February 2012. At that time, ICSD-3 [6] had not been published. The participants were already registered as members of the established internet survey company’s research panel and resided throughout Japan. The study protocol was approved by the ethics committee of the Neuropsychiatric Research Institute, Tokyo, Japan (no. 64/2011). Informed consent was obtained from all participants via the survey website.

The questionnaire included demographic information: sex, age, body mass index, residential status (living alone or with family), smoking status, and alcohol consumption. Sleep habits were evaluated as sleep duration (obtained from the Pittsburgh Sleep Quality Index [PSQI], described below), weekday bedtime, and weekday wake-up time. The questionnaire also included the Japanese version of the PSQI for assessing sleep disturbances [18,19], the Japanese version of MUPS for identifying the lifetime incidence of sleepwalking (i.e., walking or sitting up while still asleep), and current episodes of nocturnal eating behavior (i.e., waking again after falling asleep to eat) and SRED-like behavior (i.e., while asleep, eating something or preparing a meal that contains unusual or inedible ingredients, such as a combination of ice cream and cheese or dishwashing detergent instead of butter) [14,15].

Of 3904 participants who accessed the survey website, 3613 responded to the questionnaire. Participants who did not complete the questionnaire (*n* = 219) or who provided invalid answers (*n* = 47) were excluded. The responses of the remaining 3347 participants (92.6%) were included in the analyses of the present study (Figure 1).

To investigate factors associated with nocturnal eating behavior and SRED-like behavior [14,15]), we conducted logistic regression analyses for age, sex, body mass index, living alone, hypnotic medication use (three or more times per week) [18,19]), smoking status, alcohol consumption, previous and/or current sleepwalking [14,15]), typical sleep duration (categorized; see below), sleep-wake schedule (delayed/not delayed; phase delay was defined using the midpoint between sleep onset and sleep offset; 4:00 AM [20]), and subjective sleep quality (categorized; see below). Typical sleep duration and sleep-wake schedule were derived from the PSQI [18,19]. Body mass index was categorized as < 25 kg/m^2^, 25–29 kg/m^2^, and ≥ 30 kg/m^2^ [21]. Typical sleep duration was categorized as < 6 and ≥ 6 hours [22,23]. PSQI scores were grouped as < 6 and ≥ 6 points [18,19]. All variables were examined initially with univariate models, and multivariate logistic regression was conducted for all variables that were significantly correlated in univariate models to determine the main correlates controlling for confounding factors. Wald statistics were used to test the significance of odds ratios (ORs) calculated from the regression analysis. SPSS version 22 (IBM SPSS, Armonk, NY, USA) was used for analysis. Statistical significance was set at *p* < 0.05.

## 3. Results

Among the 3347 participants, 160 (4.8%) reported nocturnal eating behavior and 73 (2.2%) reported SRED-like behavior one or more times per year. Nocturnal eating behavior and SRED-like behavior were concomitant in 45 participants, but 72% of participants reporting nocturnal eating behavior and 38% of participants reporting SRED-like behavior did not overlap. Table 1 lists the characteristics of study participants.

Eight factors were significantly associated with nocturnal eating behavior in univariate logistic regression analyses: female sex, body mass index ≥ 30 kg/m^2^, current smoking, use of hypnotic medication, previous and/or current sleepwalking, sleeping < 6 hours, delayed sleep-wake schedule, and higher PSQI scores (≥ 6 points). In the multiple logistic regression model, female sex, current smoking, use of hypnotic medication, previous and/or current sleepwalking, delayed sleep-wake schedule, and higher PSQI scores (≥ 6 points) were associated (Table 2).

Five factors were associated with SRED-like behavior in univariate logistic regression analyses: body mass index of 25–29 kg/m^2^, current smoking, regular alcohol consumption, previous and/or current sleepwalking, and higher PSQI scores (≥ 6 points). In the multiple logistic regression model, current smoking, use of hypnotic medication, and previous and/or current sleepwalking were associated (Table 3).

## 4. Discussion

Thus far, sleepwalking has been considered rare in NES subjects [13]. However, we found a relatively strong association between sleepwalking and nocturnal eating behavior, although it was not as strong as the association with SRED-like behavior. This may be attributable to nocturnal eating behavior in the present study being defined as eating behavior after falling asleep or to the overlap reported by some participants between the two conditions. Of note, in this study, both nocturnal eating behavior and SRED-like behavior were associated with the use of hypnotic medication, with relatively high odds ratios. Multiple studies have reported SRED induced by hypnotic medication [24,25,26,27,28], but the evidence for hypnotic medication-induced NES is scarce [29]. but both benzodiazepines and Z-drugs can cause behavioral disinhibition [30,31], which may underlie the association between the use of hypnotic medication and nocturnal eating behavior. In addition, we found that nocturnal eating behavior but not SRED-like behavior was correlated with higher PSQI scores. Although the causal relationship is unclear, an association between insomnia and NES [7,10,32,33] should be considered.

Current smoking was also associated with both nocturnal eating behavior and SRED-like behavior. Smoking is common in eating disorders [34], and especially in bulimia and/or binge eating rather than anorexia [35,36,37]. Interestingly, NES has been reported to be more frequent in populations with bulimia and/or binge eating disorder rather than anorexia [38]. Thus, nocturnal eating behavior and binge eating may possibly be related to the pathology of addiction, including smoking. As for SRED, several cases in which smoking cessation resulted in SRED have been reported [5,39]. In addition, sleep-related smoking concomitant with NES and/or SRED has been reported [40]. Although Yahia and colleagues did not find a relationship between smoking and NES [32], smoking, as an addiction, may offer an opportunity for studying the pathology of NES and SRED.

Nocturnal eating behavior was associated in our study with the female sex. Just as eating disorders typically affect females [41,42,43], NES has been reported to be more common in females [10]. However, in both NES and SRED, sex differences in the prevalence are still controversial [5,33,44,45,46,47,48,49,50]; some studies have reported that NES was more common in males [44,45]. Thus, our findings should be confirmed in larger studies.

This is the first epidemiological study to indicate a relationship between a delay in the sleep-wake rhythm and nocturnal eating behavior. One study in mice suggested that feeding behavior affects the sleep-wake rhythm via the dopaminergic system [51]. It is still unclear whether delayed eating rhythm results in a delay in the sleep-wake rhythm or vice versa. However, nocturnal secretion of melatonin has been reported to decrease in NES [9]. Moreover, some preliminary studies have suggested that therapeutic interventions that act to consolidate the circadian rhythm, such as bright light therapy [52] and treatment with agomelatine [53,54], are effective for NES. Although these circadian-focused treatments require validation, the findings of the present study indicate that a delay in the sleep-wake rhythm possibly underlies the pathophysiology of NES.

This study has some limitations. First, the study was based on self-rating without face-to-face interviews. A further very important limiting factor is that the MUPS [14] had been developed before the publication of the ICSD-3 [6], and does not include “partial or complete loss of conscious awareness during the eating episode”. However, nocturnal eating behavior and SRED-like behavior, which were expressed as “nocturnal eating” and “sleep related eating” in MUPS, respectively, were clearly distinguished by the presence/absence of consciousness [14]. This implication emphasizes the significance of focusing on the differences between nocturnal eating behavior and SRED-like behavior in this study. As this study was an analysis of self-reported data, there is a possibility that we underestimated the prevalence of subjects with SRED-like behavior who could not recall eating behavior at night. In contrast, we may have overestimated the prevalence since we required only one incident over the preceding year. Therefore, the present study should be considered preliminary in assessing the pathophysiology of NES and SRED. The second limitation is a possible sampling bias. Generally, internet users have been reported to experience more sleep problems or shorter sleep durations [55,56]. Third, this study did not confirm any questionnaires or severity scales specific to NES for affected subjects classified only by MUPs. Lastly, we identified the use of hypnotic medication in the survey [18,19], but did not record the type or amount. Similarly, the lack of information on the quantity of alcohol consumed and number of cigarettes smoked is also a limitation.

## 5. Conclusions

The present study using MUPS suggested that a considerable number of Japanese young adults may have NES, SRED, or both. Factors associated with both nocturnal eating behavior and SRED-like behavior were smoking, sleepwalking, and use of hypnotic medication. Nocturnal eating behavior but not SRED-like behavior was associated with a delayed sleep-wake schedule and sleep disturbances, as well as with the female sex. Confirmation of the reproducibility of these findings and prospective research that includes face-to-face interviews are warranted.

## Figures and Tables

**Figure 1 jcm-09-01243-f001:**
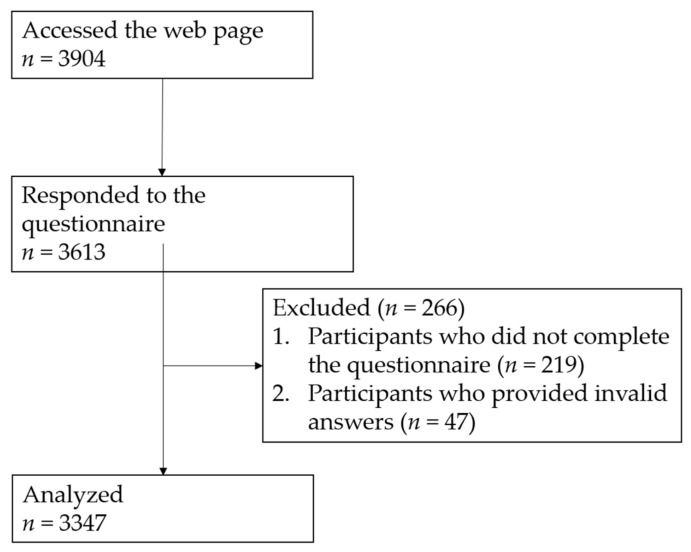
Subject flow diagram.

**Table 1 jcm-09-01243-t001:** Characteristics of study participants.

Characteristic	Total (*n* = 3347)	Nocturnal Eating Behavior (*n* = 160)	Sleep-Related Eating Disorder-Like Behavior (*n* = 73)
Age, mean (SD), year	22.9 (1.8)	22.9 (1.8)	22.8 (1.7)
Sex (percent male)	45.3	36.3	46.6
Body mass index, mean (SD), kg/m^2^	21.1 (3.6)	21.5 (4.4)	22.2 (4.4)
Current smoker (%)	10.5	20.0	21.9
Regular alcohol consumption (%)	35.1	34.4	47.9
Living alone (%)	34.1	35.0	35.6
Use of hypnotic medication (three or more times per week) (%)	2.8	13.1	12.3
Previous and/or current sleepwalking (%)	8.5	35.0	71.2
Typical sleep duration, mean (SD), hours	6.8 (1.4)	6.9 (1.6)	7.0 (1.7)
Midpoint on weekdays, mean (SD), time	4:22 (1:40)	4:41 (1:51)	4:17 (1:33)
Pittsburgh Sleep Quality Index score, mean (SD), points	5.5 (2.7)	7.6 (3.1)	7.2 (3.3)

Nocturnal eating behavior and sleep-related eating disorder-like behavior were defined as occurring at least once per year. SD, standard deviation.

**Table 2 jcm-09-01243-t002:** Factors associated with nocturnal eating behavior one or more times per year.

Predictor		Univariate Relative Risk (95% Confidence Interval) ^1^	*p*	Multivariate Relative Risk (95% Confidence Interval) ^1^	*p*
Age (years)	3347		n.s.		n.s.
Sex					
Male	1517				
Female	1830	1.485 (1.068–2.065)	<0.05	1.560 (1.103–2.206)	<0.05
Body mass index (kg/m^2^)					
<25	3016				
25–29	247		n.s.		n.s.
≥30	84	2.484 (1.219–5.062)	<0.05		n.s.
Living alone					
No	2204				
Yes	1143		n.s.		n.s.
Current smoker					
No	2995				
Yes	352	2.240 (1.495–3.356)	<0.001	1.980 (1.286–3.047)	<0.01
Regular alcohol consumption					
No	2172				
Yes	1175		n.s.		n.s.
Use of hypnotic medication (three or more times per week)					
No	3253				
Yes	94	6.445 (3.854–10.778)	<0.001	4.054 (2.306–7.129)	<0.001
Previous and/or current sleepwalking					
No	3062				
Yes	285	6.955 (4.894–9.886)	<0.001	6.249 (4.335–9.009)	<0.001
Typical sleep duration (hours)					
≥6	2723				
<6	624	1.539 (1.067–2.219)	<0.05		n.s.
Sleep-wake schedule^2^					
Not delayed	1438				
Delayed	1909	1.507 (1.078–2.107)	<0.05	1.478 (1.042–2.096)	<0.05
Pittsburgh Sleep Quality Index score (points)					
<6	1827				
≥6	1520	2.690 (1.915–3.779)	<0.001	1.871 (1.304–2.684)	<0.01

^1^ Relative risks approximated to odds ratios. ^2^ Phase delay was defined using 4:00 AM as the midpoint. n.s., not significant.

**Table 3 jcm-09-01243-t003:** Factors associated with sleep-related eating disorder-like behavior one or more times per year.

Predictor		Univariate Relative Risk (95% Confidence Interval) ^1^	*p*	Multivariate Relative Risk (95% Confidence interval) ^1^	*p*
Age (years)	3347		n.s.		n.s.
Sex					
Male	1517				
Female	1830		n.s.		n.s.
Body mass index (kg/m^2^)					
<25	3016				
25–29	247	2.115 (1.068–4.188)	<0.05		n.s.
≥30	84		n.s.		n.s.
Living alone					
No	2204				
Yes	1143		n.s.		n.s.
Current smoker					
No	2995				
Yes	352	2.454 (1.394–4.322)	<0.01	1.998 (1.072–3.724)	<0.05
Regular alcohol consumption					
No	2172				
Yes	1175	1.724 (1.083–2.744)	<0.05		n.s.
Use of hypnotic medication (three or more times per week)					
No	3253				
Yes	94	5.276 (2.542–10.951)	<0.001	3.750 (1.606–8.755)	<0.01
Previous and/or current sleepwalking					
No	3062				
Yes	285	32.318 (19.137–54.576)	<0.001	30.113 (17.764–51.044)	<0.001
Typical sleep duration (hours)					
≥6	2723				
<6	624		n.s.		n.s.
Sleep-wake schedule^2^					
Not delayed	1438				
Delayed	1909		n.s.		n.s.
Pittsburgh Sleep Quality Index (points)					
<6	1827				
≥6	1520	1.848 (1.151–2.969)	<0.05		n.s.

^1^ Relative risks approximated to odds ratios. ^2^ Phase delay was defined using 4:00 AM as the midpoint. n.s., not significant.

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
