# Peer review of "Prevalence and Associated Factors of Nocturnal Eating Behavior and Sleep-Related Eating Disorder-Like Behavior in Japanese Young Adults: Results of an Internet Survey Using Munich Parasomnia Screening"

_jcm, 2020, doi:10.3390/jcm9041243_

Round 1
Reviewer 1 Report
Comments:
Line 39, “the two disorders are distinguishable by whether there is awareness” as is now, this is a free statement and is seemingly still debated. Reference 6 is the only reference for this distinction (and is not placed after the statement from the authors). Reference 6 states that the distinction is still a controversial matter quote “At present, it is unclear whether SRED and NES constitute independent disorders, or should be lumped under the same diagnosis”.
The introduction could focus more on the distinction between the two disorders and the controversy, or references for this distinction should be included as this is a potential source of error that the reader should be made more aware of.
Method sections is very thorough and well described.
Tables 2 and 3 could contain effect sizes for the significant results as to inform the reader about the impact of each variable, and the significance of effect sizes should be discussed as well, as this will improve relevance and usability of the data.
Line 121; it´s a bit confusing that the authors use a percentage from the complete survey (1.3 %) and then compare it with a subset percentage. In general the paragraph from 121-126 are already presented in the tables and it seems more as a repeat of already mentioned data than a discussion of them, and if so they should be in the results section instead.
Limitations
If the defining differences between NES and SRED are if the participant is unable to recall their behavior, online measures such as questionnaires pose a limitation. How can it discern if the participants are unaware of their own behavior? This could increase the risk of underestimating prevalence.
Author Response
Response to Comments from Reviewer #1
Thank you very much for your valuable comments. We have revised our manuscript according to your comments as follows:
Comment #1
Line 39, “the two disorders are distinguishable by whether there is awareness” as is now, this is a free statement and is seemingly still debated. Reference 6 is the only reference for this distinction (and is not placed after the statement from the authors). Reference 6 states that the distinction is still a controversial matter quote “At present, it is unclear whether SRED and NES constitute independent disorders, or should be lumped under the same diagnosis”.
The introduction could focus more on the distinction between the two disorders and the controversy, or references for this distinction should be included as this is a potential source of error that the reader should be made more aware of.
Our response to Comment #1
As you pointed out, we should have emphasized the fact that the distinction between NES and SRED is still controversial. Accordingly, we have revised the relevant parts of the Abstract and Introduction.
Changes in the manuscript:
[Abstract]
Nocturnal (night) eating syndrome and sleep-related eating disorder have common characteristics, but are considered to differ in their level of consciousness during eating behavior and recallability.
[Introduction]
“SRED can be distinguished from NES by its unconscious eating behavior and its inability to recall, which was emphasized in the diagnostic criteria for SRED in the International Classification of Sleep Disorders, Third Edition (ICSD-3) [7]. However, since NES and SRED share several features, the distinction between the two is still controversial [8,9].”
Comment #2
Tables 2 and 3 could contain effect sizes for the significant results as to inform the reader about the impact of each variable, and the significance of effect sizes should be discussed as well, as this will improve relevance and usability of the data.
Our response to Comment #2
Unfortunately, logistic regression analysis cannot calculate the effect size. Instead, in keeping with the reviewer’s intentions, we added some discussion on the odds ratios.
Changes in the manuscript:
[Discussion]
“However, we found a relatively strong association between sleepwalking and nocturnal eating behavior, although it was not as strong as the association with SRED-like behavior.”
“Of note, in this study, both nocturnal eating behavior and SRED-like behavior were associated with the use of hypnotic medication, with relatively high odds ratios.”
Comment #3
Line 121; it’s a bit confusing that the authors use a percentage from the complete survey (1.3 %) and then compare it with a subset percentage. In general the paragraph from 121-126 are already presented in the tables and it seems more as a repeat of already mentioned data than a discussion of them, and if so they should be in the results section instead.
Our response to Comment #3
We fully agree with this comment and have removed “(1.3%)” and then added a summary in the first paragraph of the Results section.
Changes in the manuscript:
[Results]
“Nocturnal eating behavior and SRED-like behavior were concomitant in 45 participants, but 72% of participants reporting nocturnal eating behavior and 38% of participants reporting SRED-like behavior did not overlap.”
[Discussion]
We deleted the first paragraph.
Comment #4
Limitations
If the defining differences between NES and SRED are if the participant is unable to recall their behavior, online measures such as questionnaires pose a limitation. How can it discern if the participants are unaware of their own behavior? This could increase the risk of underestimating prevalence.
Our response to Comment #4
According to the reviewer’s comment, we added as a limitation that we may have underestimated the number of subjects with SRED-like behavior who could not recall their nocturnal eating behavior.
Changes in the manuscript:
[Discussion]
“As this study was an analysis of self-reported data, there is a possibility that we underestimated the prevalence of subjects with SRED-like behavior who could not recall eating behavior at night.”
Reviewer 2 Report
The authors performed a methodologically rigorous survey. Unfortunately untill now strict and universally shared diagnostic criteria for NES and SRED remain to be defined.
Probably one more limitation of the study is also the lack of a specific questionnaire on nocturnal eating for all the subjects who answered positive to the MUPS items on nocturnal eating behaviours.
Author Response
Response to Comments from Reviewer #2
Thank you very much for your valuable comments. We have revised our manuscript according to your comments as follows:
Comment #1
Probably one more limitation of the study is also the lack of a specific questionnaire on nocturnal eating for all the subjects who answered positive to the MUPS items on nocturnal eating behaviours.
Our response to Comment #1
As noted, this study did not confirm any questionnaires or severity scales specific to nocturnal eating syndrome for affected subjects and relied only on MUPS. We have added this information to the limitations paragraph.
Changes in the manuscript:
[Discussion]
“Third, this study did not confirm any questionnaires or severity scales specific to NES for affected subjects classified only by MUPs. Lastly...”
Reviewer 3 Report
This is a well written paper in a field which is not explored enough yet,
just a few considerations:
I would advise to report more clearly in the introduction that the diagnostic criteria used for SRED are those of the third edition of the ICSD and those for NES those proposed by Allison and coll. in 2010 (IJED). both the citations are already included in the references
It would be more appropriate to talk about Nocturnal Eating Behaviour and Sleep Related Eating Disorder-Like behaviour, given that the participants did not meet the diagnostic criteria for NES and SRED, even If they have symptoms of both syndromes.
Line 59 – The study was conducted in February 2012, but the authors correctly used in their analysis the diagnostic criteria for SRED of the third edition of ICSD published in 2014
Table 1 – the diagnosis was defined by a single episode in a year, which is significantly less than what necessary to diagnose SRED and NES
Line 132 - “some parasomnia traits—such as sleep drunkenness and/or behavioural disinhibition—have been considered a mechanism of both unconscious and conscious eating after bedtime [6).” This sentence is not correct, this overlap was present when SRED was diagnosed following the criteria of the second edition of the ICSD, in the third edition (the one used by the authors) the criteria have been modified and so there is no more overlap.
Line 156 – as above, it is incorrect talking of NES given the lack of several diagnostic criteria for NES .
Line 161 – Treatment with agomelatine and melatonin have given positive results in NES patients, but Reference 57 is a case report of a SRED and not of NES patient. Furthermore, given that the positive results in literature are all from the same author and only derived from case reports, even positive results in NES patients should be taken cautiously, requiring further studies to be certain.
Line 167 – “Moreover, we may have underestimated the prevalence of NES, since excessive 167 consumption of food after supper but before bedtime could not be accurately quantified. “...on the other hand, the presence of NES could be overestimated given that to be diagnosed as a NES patient, there should be a consumption of at least 25% of the daily food intake after evening meal and/or nocturnal awakening with ingestion at least twice a week for three months and not once a year.
Author Response
Response to Comments from Reviewer #3
Thank you very much for your positive comments. We have revised our manuscript according to your comments as follows:
Comment #1
I would advise to report more clearly in the introduction that the diagnostic criteria used for SRED are those of the third edition of the ICSD and those for NES those proposed by Allison and coll. in 2010 (IJED). both the citations are already included in the references
Our response to Comment #1
Following the reviewer’s advice, we added a description of the diagnostic criteria for NES and SRED to the Introduction.
Changes in the manuscript:
[Introduction]
“NES has been described as “other specified feeding or eating disorder” in the Diagnostic and Statistical Manual of Mental Disorders, Fifth Edition [1], but its criteria are relatively unclear. Therefore, the diagnostic criteria proposed by Allison and colleagues [3] have been also used.”
“SRED is classified as parasomnia in the ICSD-3 [7] and characterized by eating while “half-awake, half-asleep” or “asleep” [5,7].”
Comment #2
It would be more appropriate to talk about Nocturnal Eating Behaviour and Sleep Related Eating Disorder-Like behaviour, given that the participants did not meet the diagnostic criteria for NES and SRED, even If they have symptoms of both syndromes.
Our response to Comment #2
We agree that the wording you suggested would be more accurate in describing the subjects of this study. We therefore replaced “nocturnal eating” and “sleep-related eating” with “nocturnal eating behavior” and “SRED-like behavior”, respectively.
Changes in the manuscript:
Throughout the manuscript, we replaced “nocturnal eating” and “sleep-related eating” with “nocturnal eating behavior” and “SRED-like behavior,” respectively. We also revised the title of the manuscript: “Prevalence and associated factors of nocturnal eating behavior and sleep-related eating disorder-like behavior in Japanese young adults: results of an internet survey using Munich Parasomnia Screening.”
Comment #3
Line 59 – The study was conducted in February 2012, but the authors correctly used in their analysis the diagnostic criteria for SRED of the third edition of ICSD published in 2014
Our response to Comment #3
As you point out, ICSD-2 was used in 2012 when this study was initiated; ICSD-2, unlike ICSD-3, does not include “partial or complete loss of conscious awareness during the eating episode” in its diagnostic criteria for SRED, and the difference between NES and SRED was not clear. In addition, MUPS was created before the publication of the ICSD-3. However, nocturnal eating behavior and SRED-like behavior, which were termed “nocturnal eating” and “sleep-related eating” in MUPS, respectively, were clearly distinguished by the presence/absence of consciousness. Therefore, we think that SRED-like behavior in this study is compatible with SRED in the ICSD-3. However, for accuracy, we emphasized that ICSD-3 had not been published at the time of the investigation. In addition, we thought that it was also very important to note the differences between the definitions in ICSD-2 and ICSD-3, so we added this information to the limitations paragraph.
Changes in the manuscript:
[Experimental Section]
“At that time, ICSD-3 [7] had not been published.”
[Discussion]
“A further very important limiting factor is that the MUPS [15] had been developed before the publication of the ICSD-3 [7], and does not include “partial or complete loss of conscious awareness during the eating episode”. However, nocturnal eating behavior and SRED-like behavior, which were expressed as “nocturnal eating” and “sleep related eating” in MUPS, respectively, were clearly distinguished by the presence/absence of consciousness [15]. This implication emphasizes the significance of focusing on the differences between nocturnal eating behavior and SRED-like behavior in this study.”
Comment #4
Table 1 – the diagnosis was defined by a single episode in a year, which is significantly less than what necessary to diagnose SRED and NES
Our response to Comment #4
The reviewer’s comment is correct; we have added this information to the limitations paragraph.
Changes in the manuscript:
[Discussion]
In contrast, we may have overestimated the prevalence since we required only one incident over the preceding year. Therefore, the present study should be considered preliminary in assessing the pathophysiology of NES and SRED.
Comment #5
Line 132 - “some parasomnia traits—such as sleep drunkenness and/or behavioural disinhibition—have been considered a mechanism of both unconscious and conscious eating after bedtime [6).” This sentence is not correct, this overlap was present when SRED was diagnosed following the criteria of the second edition of the ICSD, in the third edition (the one used by the authors) the criteria have been modified and so there is no more overlap.
Our response to Comment #5
As the reviewer pointed out, we made an incorrect statement. We modified this part of the discussion to state that the overlap we recorded may be attributable to the similarity between nocturnal eating behavior and SRED-like behavior.
Changes in the manuscript:
[Discussion]
“This may be attributable to nocturnal eating behavior in the present study being defined as eating behavior after falling asleep or to the overlap reported by some participants between the two conditions.”
Comment #6
Line 156 – as above, it is incorrect talking of NES given the lack of several diagnostic criteria for NES.
Our response to Comment #6
As you pointed out, the term “NES” was incorrect, so we rewrote it as nocturnal eating behavior.
Changes in the manuscript:
[Discussion]
“This is the first epidemiological study to indicate a relationship between a delay in the sleep-wake rhythm and nocturnal eating behavior.”
Comment #7
Line 161 – Treatment with agomelatine and melatonin have given positive results in NES patients, but Reference 57 is a case report of a SRED and not of NES patient. Furthermore, given that the positive results in literature are all from the same author and only derived from case reports, even positive results in NES patients should be taken cautiously, requiring further studies to be certain.
Our response to Comment #7
Thank you for pointing out our mistake. That reference was inappropriate and has been removed. The text has been revised to emphasize that both of these treatments should be validated in future studies.
[Discussion]
“Moreover, some preliminary studies have suggested that therapeutic interventions that act to consolidate the circadian rhythm, such as bright light therapy [53] and treatment with agomelatine [54,55], are effective for NES. Although these circadian-focused treatments require validation, the findings of the present study indicate that a delay in the sleep-wake rhythm possibly underlies the pathophysiology of NES.”
Comment #8
Line 167 – “Moreover, we may have underestimated the prevalence of NES, since excessive 167 consumption of food after supper but before bedtime could not be accurately quantified. “...on the other hand, the presence of NES could be overestimated given that to be diagnosed as a NES patient, there should be a consumption of at least 25% of the daily food intake after evening meal and/or nocturnal awakening with ingestion at least twice a week for three months and not once a year.
Our response to Comment #8
We regret this oversight. As you pointed out, we might have overestimated both NES and SRED since we used the 1-year prevalence of nocturnal eating behavior and SRED-like behavior. We therefore deleted the paragraph describing this possibility and revised the limitations paragraph.
Changes to the manuscript:
[Discussion]
In contrast, we may have overestimated the prevalence since we required only one incident over the preceding year. Therefore, the present study should be considered preliminary in assessing the pathophysiology of NES and SRED.
Round 2
Reviewer 2 Report
None